# Effects of Muscle Fiber Composition on Meat Quality, Flavor Characteristics, and Nutritional Traits in Lamb

**DOI:** 10.3390/foods14132309

**Published:** 2025-06-29

**Authors:** Yu Fu, Yang Chen, Xuewen Han, Dandan Tan, Jinlin Chen, Cuiyu Lai, Xiaofan Yang, Xuesong Shan, Luiz H. P. Silva, Huaizhi Jiang

**Affiliations:** 1College of Animal Science and Technology, Jilin Agricultural University, Changchun 130118, China; 15164460626@163.com (Y.F.); chenyang7419@163.com (Y.C.); hanxuewen1123@163.com (X.H.); 13850139585@163.com (D.T.); chenjinlin7314@163.com (J.C.); 19843927673@163.com (C.L.); yangxiaofan1117@163.com (X.Y.); jianghz6806@126.com (H.J.); 2Department of Agriculture and Food Science, Western Kentucky University, Bowling Green, KY 42101-1066, USA

**Keywords:** muscle fiber type, fatty acid, amino acid, volatile flavor compounds, metabolomics

## Abstract

Skeletal muscle fiber type composition critically influences lamb meat quality. This study examined the relationships between muscle fiber types and key quality traits, including tenderness, color, lipid and amino acid profiles, and volatile flavor compounds. MyHC I (slow-twitch oxidative fibers) positively correlated with desirable traits such as increased redness, water-holding capacity, unsaturated fatty acids, and essential amino acids. Conversely, MyHC IIb (fast glycolytic fibers) was linked to reduced tenderness and higher levels of off-flavor compounds. MyHC IIa and IIx showed minimal effects. Untargeted metabolomics comparing muscles with high versus low slow-twitch fiber proportions revealed differential metabolites enriched in sphingolipid and arginine-proline metabolism pathways. These results suggest that a higher proportion of oxidative fibers enhances both the sensory and nutritional qualities of lamb meat by modulating lipid metabolism, amino acid availability, and flavor formation.

## 1. Introduction

Lamb meat is highly valued in global markets not only for its palatability but also for its rich nutritional profile. As a source of high-quality protein, lamb supplies all essential amino acids required for human growth and development. It is also a key dietary source of micronutrients such as iron (especially heme iron, which is highly bioavailable), zinc, selenium, and B vitamins including B12 and niacin [1,2]. Moreover, lamb contains a favorable balance of saturated and unsaturated fatty acids, including conjugated linoleic acid (CLA) and omega-3 polyunsaturated fats, which have been linked to cardiovascular and metabolic health benefits [3]. As such, improving lamb meat quality is not only vital for meeting consumer sensory expectations but also holds broader implications for human nutrition and public health [4].

Meat quality is a multifactorial trait shaped by genetic, physiological, and biochemical characteristics of skeletal muscle. Among these, the composition of muscle fiber types has emerged as a key determinant, influencing not only the appearance and texture of meat but also its nutritional value and flavor profile [5,6]. In livestock species such as cattle, pigs, and sheep, skeletal muscle is composed of diverse fiber types classified according to their contractile speed and metabolic specialization. These include slow-twitch oxidative fibers (type I) and fast-twitch fibers, which can be further subdivided into oxidative (type IIa), intermediate (type IIx), and glycolytic (type IIb) types [7]. The expression of specific myosin heavy chain (MyHC) isoforms serves as a molecular marker for each fiber type and reflects the muscle’s adaptation to different functional demands [8].

Meat quality assessment encompasses a suite of physical, chemical, and sensory parameters. These include tenderness, juiciness, flavor, and color, along with quantitative traits such as pH, water-holding capacity, cooking loss, and shear force. From a compositional standpoint, quality is shaped by intramuscular fat (IMF) content and the profiles of amino acids, fatty acids, and low-molecular-weight metabolites such as nucleotides and sugars [9,10]. These compounds directly affect taste and aroma and are, in turn, modulated by muscle fiber composition [11].

In ruminants, the *longissimus lumborum* (*LL*) muscle is widely used as a benchmark for meat quality due to its commercial significance and sensitivity to muscle fiber type. Muscles enriched in oxidative fibers (MyHC I) tend to display superior water-holding capacity, deeper red color, slower pH decline postmortem, and improved tenderness—traits linked to their aerobic metabolism and high mitochondrial density [12]. In contrast, glycolytic fiber-rich muscles (MyHC IIb) are more susceptible to rapid pH decline, increased drip and cooking losses, and inferior tenderness, due to their anaerobic metabolism and limited oxidative capacity [13].

The biochemical underpinnings of these differences lie in the enzymatic and metabolic properties of the fiber types. Oxidative fibers (types I and IIa) are characterized by the elevated activity of mitochondrial enzymes such as succinate dehydrogenase (SDH) and citrate synthase (CS), enabling sustained aerobic respiration via β-oxidation and the tricarboxylic acid (TCA) cycle [14,15]. These fibers are rich in myoglobin and mitochondria, contributing to their dark color and metabolic resilience. Glycolytic fibers (type IIb), by contrast, rely on anaerobic glycolysis, with a high activity of enzymes such as lactate dehydrogenase (LDH) and phosphofructokinase (PFK), leading to rapid ATP generation but also to lactate accumulation and faster postmortem acidification [16]. These enzymatic distinctions not only shape muscle energetics but also critically affect the biochemical pathways that govern meat color, pH, and tenderness during and after slaughter [17].

Beyond physical traits, muscle fiber types also influence the biochemical flavor landscape of meat. Key sensory properties such as taste and aroma arise from the interplay between taste-active compounds (e.g., free amino acids, peptides, nucleotides) and volatile aroma compounds formed during cooking. Notably, branched-chain fatty acid derivatives such as 4-methyloctanoic acid and 4-methylnonanoic acid contribute to the distinct “mutton flavor” often perceived negatively by consumers in certain markets [18,19]. The abundance of such compounds is influenced by both fiber metabolism and lipid composition [20].

Although the role of muscle fiber composition in shaping meat quality is well established in pigs and cattle, integrated studies in lamb that link MyHC isoform expression with biochemical and sensory quality traits remain scarce. In particular, relationships between fiber type distribution and volatile compounds, fatty acid and amino acid profiles, and global metabolomic changes are poorly characterized.

In this study, we sought to systematically characterize the associations between MyHC isoform expression and key meat quality attributes in lamb *longissimus lumborum* muscle. Using a multi-omics approach—including qRT-PCR, chemical composition analysis, gas chromatography–mass spectrometry (GC–MS), and untargeted metabolomics—we investigated how fiber type composition influences traditional meat quality metrics, nutritional composition, and flavor-related metabolites. Our findings provide new insights into the molecular basis of lamb meat quality and offer a foundation for improving eating quality through genetic selection, nutritional modulation, or production management.

## 2. Materials and Methods

### 2.1. Animals and Sample Collection

A total of 30 healthy 6-month-old intact male lambs (F1 crossbred of Suffolk rams and Small-tailed Han ewes) were selected from a commercial farm. All animals were housed indoors under consistent management conditions, with ad libitum access to feed and water. The diet was formulated to meet the nutritional requirements for growing lambs, and the detailed feed composition and nutrient content are provided in Table 1. Prior to slaughter, all animals were fasted for 24 h with free access to water. Slaughter was conducted in accordance with institutional animal care and ethical guidelines. Within 30 min postmortem, the *longissimus lumborum* (*LL*) muscle was collected, immediately snap-frozen in liquid nitrogen, and stored at −80 °C for subsequent molecular and biochemical analyses.

### 2.2. Analysis of Muscle Fiber Type Marker Gene Expression

Total RNA was extracted from *longissimus lumborum* (*LL*) muscle samples using TRIzol reagent (R0016, Beyotime, Shanghai, China), following the manufacturer’s protocol. RNA concentration and purity were assessed using a NanoDrop spectrophotometer (Thermo Fisher Scientific, Waltham, MA, USA). First-strand cDNA was synthesized from 1 μg of total RNA using a BeyoRT™II First Strand cDNA Synthesis Kit (D7168M, Beyotime).

Quantitative real-time PCR (qPCR) was conducted using SYBR Green Master Mix (D7260, Beyotime, Shanghai, China)) on a Real-Time PCR System (CFX96 opus, Bio-Rad, Hercules, CA, USA). Expression levels of four muscle fiber subtype maker genes—MyHC I, IIa, IIx, and IIb—were quantified, with β-actin serving as the internal control. Relative expression levels were calculated using the 2^−ΔΔCt^ method. Each sample was analyzed in triplicate technical replicates to ensure accuracy and reproducibility. Primer sequences used in the analysis are provided in Table 2.

### 2.3. Measurement of Meat Quality Traits

Meat quality traits and muscle composition were evaluated using thirty *longissimus lumborum* (*LL*) samples. The ultimate pH (pH_24_) of the *longissimus lumborum* muscle was determined 24 h postmortem using a calibrated pH meter (S220, Mettler Toledo, Columbus, OH, USA) equipped with a spear-tip probe, which was directly inserted into the muscle sample. For each sample, pH was measured at three different points, and the average value was used for analysis. Meat color was assessed after 20 min of blooming at room temperature to allow oxygenation of myoglobin. A colorimeter (CR-400, Konica Minolta, Tokyo, Japan) was used to record lightness (L*), redness (a*), and yellowness (b*) according to the CIE system. Measurements were taken with an 8 mm aperture, 10° standard observer, and D65 illuminant. The device was operated in tristimulus mode, and spectral reflectance data were not recorded. Cooking loss was determined by weighing 40 g meat samples before and after thermal treatment in a water bath at 75 °C until the internal temperature reached 70 °C. Meat tenderness was measured using a texture analyzer (TA.XTplus, Stable Micro Systems, Godalming, UK) equipped with a Warner–Bratzler shear force (WBSF) blade. Each cooked meat sample was cut into 1.0 cm × 1.0 cm × 3.0 cm strips parallel to the muscle fiber direction. Shear force measurements were performed perpendicularly to the fiber direction at a crosshead speed of 200 mm/min. The peak force required to shear through the sample was recorded, and the average of three replicates was used as the final value. Muscle moisture, crude protein, and intramuscular fat contents were determined using standard methods: direct drying (GB 5009.3-2016) [21], Kjeldahl nitrogen determination (GB 5009.5-2016) [22], and Soxhlet ether extraction (GB 5009.6-2016) [23], respectively.

### 2.4. Fatty Acid Profile Analysis

Lipid extraction from muscle samples was performed using the Folch method, followed by methylation to form fatty acid methyl esters (FAMEs). FAMEs were then analyzed by gas chromatography (GC) equipped with a flame ionization detector. Individual fatty acids were identified and quantified by comparing retention times with those of known standards. The analysis was conducted in accordance with the Chinese National Food Safety Standard GB 5009.168-2016 [24].

### 2.5. Free Amino Acid Analysis

Free amino acids were extracted from homogenized and deproteinized muscle samples and quantified using a high-speed amino acid analyzer. The concentrations of total and individual amino acids, including essential amino acids, were measured and compared among samples. The procedure was performed following the Chinese National Food Safety Standard GB 5009.124-2016 [25].

### 2.6. Volatile Flavor Compound Analysis

Approximately 5 g of *longissimus lumborum* muscle tissue was pretreated and subjected to analysis of volatile flavor compounds using gas chromatography–mass spectrometry (GC-MS). Seven volatile compounds associated with the characteristic odor of lamb (including 4-methyloctanoic acid, 4-methylnonanoic acid, hexanal, benzaldehyde, pentanol, decadienal, and nonanal) were identified. Volatile compounds were extracted using solid-phase microextraction (SPME), and identification was performed by comparing retention times and mass spectra with those in commercial libraries (e.g., NIST).

### 2.7. Immunofluorescence Staining of Muscle Fiber Types

Muscle fiber type composition was determined using immunofluorescence staining. Serial cryosections (10 µm thick) of the LL muscle were fixed, blocked, and incubated with monoclonal antibodies specific for slow-twitch (MyHC I, ab234431, Abcam, Cambridge, UK) and fast-twitch (MyHC II, ab91506, Abcam, Cambridge, UK) fibers. Secondary antibodies conjugated with fluorophores were applied, and images were captured using a fluorescence microscope (IX73, Olympus, Tokyo, Japan). The relative proportion of slow- and fast-twitch fibers was quantified using ImageJ software (v1.54g). Based on the proportion of slow-twitch fibers, six samples with the highest and six with the lowest MyHC I content were selected for metabolomic analysis.

### 2.8. Untargeted Metabolomics

Untargeted metabolomic profiling of *longissimus lumborum* muscle was performed using UHPLC-QTOF-MS. Briefly, frozen samples were thawed at 4 °C, homogenized, and extracted with methanol/acetonitrile/water (2:2:1, *v*/*v*). After vortexing, sonication, and centrifugation, the supernatant was dried and reconstituted in acetonitrile/water (1:1, *v*/*v*) for analysis. Chromatographic separation was conducted on an Agilent 1290 UHPLC system (Agilent Technologies, Santa Clara, CA, USA) using a HILIC column, and metabolites were analyzed on a Triple TOF 6600 mass spectrometer (AB SCIEX, Framingham, MA, USA), in both positive and negative ion modes. The MS was operated in data-dependent acquisition mode with dynamic exclusion. Raw data were converted to mzXML format and processed using XCMS for peak detection and alignment. Metabolite annotation was performed by matching MS/MS spectra against public databases (HMDB, KEGG, METLIN) and an in-house spectral library based on accurate mass and fragmentation patterns. Principal component analysis (PCA), heatmap clustering, volcano plots, and pathway enrichment were performed using MetaboAnalyst 5.0 [26]. Metabolites with VIP > 1, *p* < 0.05, and |log2FC| > 1 were considered significantly different. Pathway enrichment was performed based on KEGG database annotations.

### 2.9. Statistical Analysis

Correlation analyses between MyHC isoform expression and meat quality traits, fatty acid profiles, amino acids, and volatile compounds were performed using Pearson’s correlation coefficients in SPSS 27.0 software. Differences between high and low slow-twitch groups were analyzed using unpaired *t*-tests. Statistical significance was set at *p* < 0.05, and highly significant differences at *p* < 0.01. Graphs were generated using GraphPad Prism 9.5.

## 3. Results

### 3.1. Association Between Muscle Fiber Subtypes and Meat Quality Traits

To evaluate how muscle fiber type composition affects lamb meat quality, we analyzed the correlations between the expression levels of four MyHC isoforms (MyHC I, IIa, IIx, and IIb) (Table 3) and key meat quality parameters (Table 4).

MyHC I expression was significantly positively correlated with meat redness (a*, *p* < 0.05) and moisture content (*p* < 0.05) and negatively correlated with intramuscular fat content (*p* < 0.05). In contrast, MyHC IIb expression was positively associated with shear force (*p* < 0.05) and ultimate pH (*p* < 0.01), suggesting reduced tenderness, and negatively correlated with meat lightness (L*, *p* < 0.05), yellowness (b*, *p* < 0.05), and cooking loss (*p* < 0.05). Collectively, these data indicated that a higher proportion of slow oxidative fibers (MyHC I) contributes to redder, juicier meat with improved water-holding capacity, while a predominance of fast glycolytic fibers (MyHC IIb) is associated with a tougher texture and less desirable visual and moisture-related traits (Figure 1).

### 3.2. Relationship Between MyHC Isoforms and Muscle Lipid Composition

To further explore the relationship between muscle fiber composition and intramuscular lipid metabolism, we quantified the relative proportions of major fatty acids in the *longissimus lumborum* muscle. Descriptive statistics of fatty acid composition are presented in Table 5.

Correlation analysis revealed significant associations between MyHC isoform expression and intramuscular fatty acid profiles (Figure 2). Specifically, MyHC I expression was positively correlated with unsaturated fatty acids (UFAs), including oleic acid (C18:1n-6, *p* < 0.01) and total UFAs (*p* < 0.05), and negatively correlated with saturated fatty acids (SFAs, *p* < 0.05), such as myristic acid (C14:0, *p* < 0.05) and palmitic acid (*p* < 0.05). These findings suggest that a higher proportion of slow oxidative fibers is associated with a more favorable, health-promoting lipid profile in lamb meat.

In contrast, MyHC IIb expression exhibited an opposing trend, although the correlations were not statistically significant for all fatty acid types. MyHC IIa and IIx showed only weak or nonsignificant correlations with lipid parameters, implying a limited role in modulating fatty acid composition under the current feeding conditions.

### 3.3. Association Between MyHC Isoforms and Free Amino Acid Profiles

To further investigate how muscle fiber types influence the accumulation of free amino acids—key contributors to both meat flavor and nutritional value—we quantified the concentrations of major amino acids in the *longissimus lumborum* muscle. Descriptive statistics of amino acid contents are presented in Table 6.

Correlation analysis revealed that MyHC I expression was significantly positively associated with several flavor-enhancing and nutritionally important amino acids, including glutamic acid, alanine, valine, and lysine (all *p* < 0.01), as well as total essential amino acids (EAA, *p* < 0.05). These results suggest that muscles with a higher proportion of slow-twitch oxidative fibers tend to accumulate more amino acids that contribute to both palatability and nutritional quality.

In contrast, MyHC IIb expression showed negative correlations with most amino acids, aligning with its previously observed association with lower overall meat quality. MyHC IIa and IIx isoforms exhibited limited or inconsistent associations with amino acid concentrations, indicating a lesser role in determining flavor-related traits under the current production conditions. The correlation patterns between MyHC isoforms and amino acid profiles are visualized in Figure 3.

### 3.4. Correlation Between Muscle Fiber Types and Volatile Flavor Compounds

To examine the potential influence of muscle fiber types on meat aroma, we quantified the concentrations of key volatile flavor compounds in the *longissimus lumborum* muscle using GC–MS. Summary statistics of detected compounds are provided in Table 7.

A correlation heatmap analysis revealed distinct associations between MyHC isoform expression and specific volatile compounds (Figure 4). MyHC I expression was positively correlated with nonanal, a lipid-derived aldehyde that contributes to desirable fatty and citrus-like aromas. In contrast, MyHC IIb and IIx isoforms were positively associated with 4-methyloctanoic acid and 4-methylnonanoic acid, two branched-chain fatty acid derivatives responsible for the characteristic “mutton flavor”, which can be perceived as undesirable in some consumer markets.

Interestingly, MyHC IIx expression showed a significant negative correlation with 1-penten-3-ol (*p* < 0.05), a compound known to impart grassy and green off-flavor notes, suggesting that higher IIx expression may reduce the accumulation of undesirable aroma volatiles. In contrast, MyHC IIa exhibited a significant positive correlation with 1-penten-3-ol (*p* < 0.01), indicating that this intermediate fiber type may contribute to the generation of such off-flavor compounds. These findings highlight the differential roles of intermediate muscle fiber types (IIa and IIx) in modulating the volatile metabolite profile of lamb meat and suggest that, even within non-extreme fiber types, subtle variations in expression may significantly influence meat flavor characteristics.

### 3.5. Muscle Fiber Typing Validates Grouping for Metabolomic Analysis

To validate the classification of samples used in the metabolomic analysis, we quantified the proportion of slow-twitch (type I) and fast-twitch (type II) muscle fibers in the *longissimus lumborum* muscle of 30 lambs using immunofluorescence staining. Based on previous findings indicating that MyHC I expression is positively associated with superior meat quality traits, we selected six animals with the highest and six with the lowest proportions of slow-twitch fibers for metabolomic profiling (Figure 5). A bar graph comparison revealed a highly significant difference (*p* < 0.001) in the percentage of type I fibers between the two groups, confirming the effectiveness and reliability of the grouping strategy for downstream analysis.

### 3.6. Metabolomic Differences Between High and Low MyHC I Groups

To elucidate the metabolic differences associated with muscle fiber composition, an untargeted metabolomic analysis was performed on the *longissimus lumborum* samples from lambs with high (H) and low (L) proportions of MyHC I (slow-twitch) fibers.

A total of 993 metabolites were identified under both positive and negative ionization modes. The hierarchical clustering heatmap (Figure 6A) shows a clear separation between the H and L groups, with distinct clusters of upregulated and downregulated metabolites, indicating substantial metabolic reprogramming associated with fiber type composition.

Principal component analysis (PCA, Figure 6B) revealed that the first principal component (PC1) accounted for 57.8% of the variance, clearly separating the H and L groups with minimal intra-group variation. This confirmed that metabolic profiles are strongly influenced by muscle fiber type.

The volcano plot (Figure 6C) identified 27 significantly altered metabolites (Table 8). The high MyHC I group showed increased levels of metabolites involved in lipid metabolism (e.g., C17-sphinganine, isobutyryl-l-carnitine), antioxidant defense (e.g., juglone, evodiamine), and peptides and amino acid derivatives (e.g., gamma-glutamylvaline, Lys-Trp-Arg). In contrast, phosphocreatine, Met-Phe, and Thr-His were downregulated in this group, reflecting lower glycolytic activity and proteolysis. These differences reflected alterations in carbohydrate metabolism, membrane phospholipid turnover, and redox regulation.

Metabolite set enrichment analysis (Figure 6D) revealed that beta-alanine metabolism, sphingolipid metabolism, arginine and proline metabolism, and arachidonic acid metabolism were enriched to varying degrees. Among these, beta-alanine metabolism showed the highest enrichment ratio and lowest *p*-value, indicating a potentially important role in regulating intracellular pH buffering and muscle fatigue resistance. Sphingolipid metabolism and arginine and proline metabolism, both associated with membrane stability, oxidative stress modulation, and collagen synthesis, were also significantly enriched, suggesting enhanced structural maintenance and redox homeostasis in slow-twitch muscle fibers. Additionally, arachidonic acid metabolism, though less enriched, may be implicated in inflammatory signaling and membrane phospholipid turnover. These enriched pathways collectively indicated a metabolic orientation of slow-twitch fibers toward sustaining oxidative capacity, structural integrity, and anti-fatigue potential.

Collectively, these results support the conclusion that muscle fiber type composition significantly impacts the muscle’s metabolic landscape, potentially contributing to differences in meat flavor, nutritional quality, and biochemical stability.

## 4. Discussion

Skeletal muscle fiber type is a fundamental determinant of meat quality, influencing both technological traits and sensory attributes. Although previous studies highlighted the general importance of fiber type composition in shaping meat quality in species such as pigs and cattle [5,27], they have not systematically examined the specific roles of individual MyHC isoforms in lamb, particularly with respect to volatile flavor compounds and metabolic pathways.

Our study bridges this gap by comprehensively analyzing the expression of four major MyHC isoforms (I, IIa, IIx, and IIb) in lamb *longissimus lumborum* muscle and linking them to a broad range of meat quality traits, including color parameters, shear force, cooking loss, amino acid composition, fatty acid profiles, volatile flavor compounds, and untargeted metabolomic features. This multi-dimensional analysis allowed us to dissect the metabolic and sensory consequences of each fiber type with unprecedented resolution in lamb, a species for which detailed fiber–flavor relationships have been poorly characterized.

We found that MyHC I (oxidative slow-twitch fibers) was positively associated with favorable meat traits, such as increased redness (a*), water-holding capacity, unsaturated fatty acids (e.g., oleic acid), and essential amino acids. These results are consistent with previous findings in pigs and yaks, where oxidative fibers are linked to improved color stability and nutritional value [12,28]. The increased redness in MyHC I-rich muscles likely results from an elevated myoglobin content and reduced glycolytic activity, which mitigates pH decline postmortem and preserves meat color [29].

The observed enrichment of unsaturated fatty acids (UFAs) in oxidative fibers can be attributed to their distinct metabolic properties. Oxidative fibers, characterized by MyHC I expression, possess greater mitochondrial density, a higher capillary supply, and elevated enzymatic activities associated with fatty acid transport and β-oxidation [30]. These features facilitate the preferential uptake and catabolism of UFAs, especially oleic acid, for sustained aerobic energy production. This physiological preference helps explain why MyHC I-rich muscles accumulate more UFAs, contributing to their nutritional advantages and flavor precursor potential [31,32].

Furthermore, the positive association between MyHC I and essential amino acids such as lysine, valine, and glutamic acid suggests enhanced nutritional quality and flavor precursors, which is in line with observations in pigs, where oxidative muscles contain more taste-active amino acids and dipeptides [33]. Conversely, MyHC IIb fibers, representative of fast-glycolytic fibers, showed negative associations with amino acid content and were linked to higher shear force and greater cooking loss, indicating tougher and less juicy meat. This supports previous findings that glycolytic fibers are prone to excessive postmortem lactate accumulation, leading to adverse effects on proteolysis and tenderness [18,34].

Volatile compound profiling further revealed that muscle fiber types significantly modulate lamb flavor. Despite not reaching statistical significance, MyHC I expression showed a strong positive correlation with nonanal, a pleasant aldehyde generated from oleic acid oxidation, known to impart fatty and citrus-like aromas [35]. The enrichment of oleic acid in oxidative fibers provides a plausible precursor source for nonanal, highlighting the contribution of fiber metabolism to flavor generation. In contrast, MyHC IIb and IIx were strongly associated with 4-methyloctanoic acid and 4-methylnonanoic acid, branched-chain fatty acids that are key contributors to the species-specific “mutton” odor often rejected by consumers [36]. These volatiles are thought to derive from branched-chain amino acid catabolism and are more prevalent in glycolytic muscle regions, as previously reported in sheep and goats [37].

In addition, our study provides novel evidence that MyHC IIx expression exhibited a significant negative correlation with 1-penten-3-ol (*p* < 0.05), a compound linked to grassy and green off-flavor notes, suggesting that higher IIx expression may reduce the accumulation of unfavorable aroma volatiles. MyHC IIa, often considered an intermediate fiber type, demonstrated a significant positive correlation with 1-penten-3-ol (*p* < 0.01), indicating a differential effect between intermediate fiber types. These findings highlight the potential role of muscle fiber type composition in shaping the flavor profile of lamb through the modulation of volatile metabolite production and underscore the influence of intermediate fibers on flavor quality—an area that warrants further investigation [38].

Untargeted metabolomics further supported the metabolic divergence among fiber types. Muscles with high MyHC I expression exhibited increased levels of C17-sphinganine and isobutyryl-L-carnitine, indicative of enhanced mitochondrial lipid β-oxidation [39]. The upregulation of carnitine-related metabolites aligns with the aerobic metabolic nature of oxidative fibers. Moreover, these muscles also showed elevated levels of γ-glutamyl peptides and plant-derived antioxidants (e.g., evodiamine, juglone), suggesting improved flavor complexity, kokumi taste potential, and oxidative stability. The γ-glutamyl peptides were associated with mouthfulness and prolonged flavor perception in meat [40], while the antioxidant compounds may support lipid stability and shelf life [41]. Their selective enrichment in oxidative fibers may reflect differences in uptake or metabolic utilization.

In contrast, the lower levels of phosphocreatine and glycolytic dipeptides (e.g., Met-Phe, Thr-His) in oxidative muscles are consistent with their reduced reliance on anaerobic energy systems. These metabolites in fast-twitch glycolytic fibers favor the utilization of rapid ATP generation through glycolysis and phosphagen pathways [42]. This divergence in energy metabolism provides a molecular basis for the observed meat quality differences between oxidative and glycolytic fiber types.

Overall, our findings substantially expand upon prior research by providing an integrative view of how MyHC isoform composition influences not only structural and nutritional traits but also flavor compound profiles and underlying metabolic mechanisms in lamb. Unlike earlier studies that typically classified muscle fibers into broad oxidative or glycolytic categories, our work offers a detailed characterization of all four major MyHC isoforms (I, IIa, IIx, IIb), allowing for more precise associations with specific meat quality parameters. By simultaneously evaluating a wide range of meat quality traits—including volatile flavor compounds, metabolomic markers, and conventional technological indices—our study establishes comprehensive links between fiber type composition and both sensory and nutritional attributes. Furthermore, by focusing specifically on lamb meat, our research fills a critical gap in the literature, as high-resolution data on muscle fiber–flavor interactions in this species have been limited. This species-specific insight represents a valuable contribution to the understanding of lamb meat quality and consumer acceptability.

These findings provide important implications for meat production and quality control. Increasing the proportion of oxidative fibers (especially MyHC I) through dietary strategies, exercise, or selective breeding may improve consumer acceptance of lamb meat by enhancing its tenderness, nutritional value, and flavor profile while reducing the intensity of undesirable mutton odor. Given the unique challenges of lamb flavor perception in certain markets, our study suggests that muscle fiber modulation could serve as a viable strategy to develop lamb products with more favorable sensory properties.

In conclusion, this study provides a comprehensive framework linking MyHC isoform expression to lamb meat quality, combining molecular, metabolic, and sensory data to reveal how muscle fiber phenotype shapes flavor, tenderness, and nutritional composition. Future research incorporating single-cell transcriptomics, fiber-specific proteomics, and targeted flavoromics will further elucidate the regulatory networks underlying muscle fiber–meat quality interactions and help design optimized production strategies for high-quality lamb meat.

## 5. Conclusions

This study demonstrates that MyHC I expression serves as a key biomarker of superior meat quality, being positively associated with favorable color, water retention, unsaturated fatty acids, essential amino acids, and pleasant flavor compounds. In contrast, MyHC IIb expression correlates with undesirable texture and the accumulation of volatile compounds responsible for strong species-specific odors. Metabolomic analysis further reveals that oxidative muscle fibers support lipid oxidation pathways and redox balance conducive to improved sensory and nutritional meat traits. These results suggest that promoting oxidative fiber development—via genetic, nutritional, or management interventions—may be an effective strategy to enhance the eating quality, flavor acceptability, and overall value of lamb meat.

## Figures and Tables

**Figure 1 foods-14-02309-f001:**
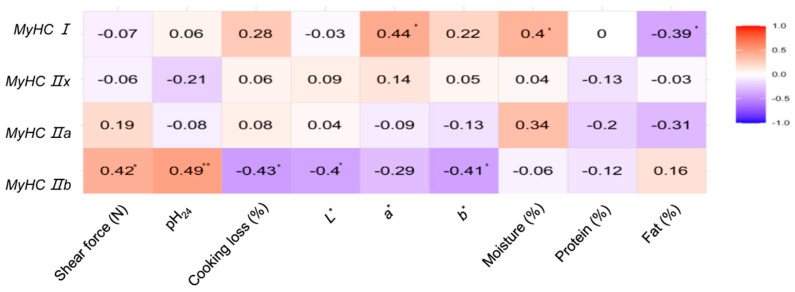
Association between MyHC isoforms and meat quality traits. The intensity of the color corresponds to the magnitude of the correlation coefficient (r). Significance levels are indicated by asterisks: * *p* < 0.05; ** *p* < 0.01.

**Figure 2 foods-14-02309-f002:**
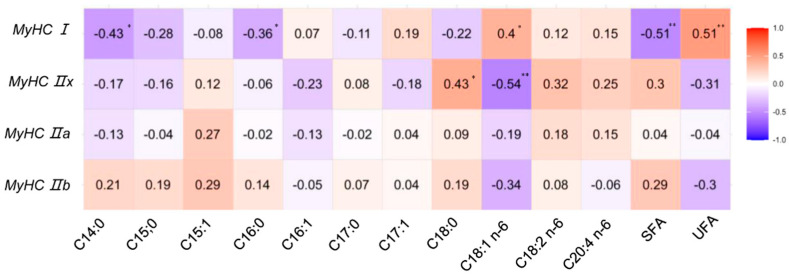
Relationship between MyHC isoforms and muscle lipid composition. The intensity of the color corresponds to the magnitude of the correlation coefficient (r). Significance levels are indicated by asterisks: * *p* < 0.05; ** *p* < 0.01.

**Figure 3 foods-14-02309-f003:**
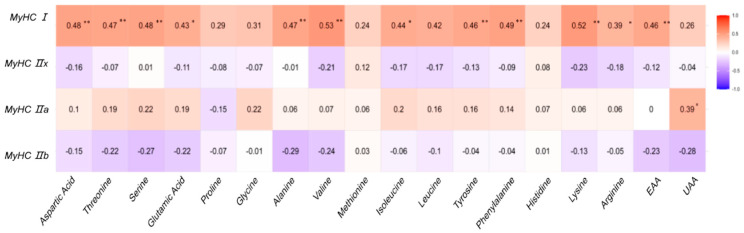
Association between MyHC isoforms and free amino acid profiles. The intensity of the color corresponds to the magnitude of the correlation coefficient (r). Significance levels are indicated by asterisks: * *p* < 0.05; ** *p* < 0.01.

**Figure 4 foods-14-02309-f004:**
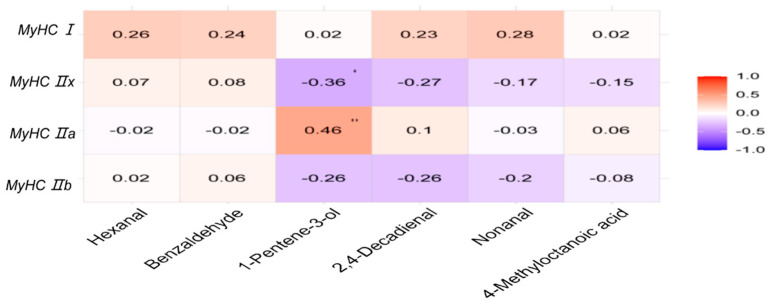
Correlation between muscle fiber types and volatile flavor compounds. The intensity of the color corresponds to the magnitude of the correlation coefficient (r). Significance levels are indicated by asterisks: * *p* < 0.05; ** *p* < 0.01.

**Figure 5 foods-14-02309-f005:**
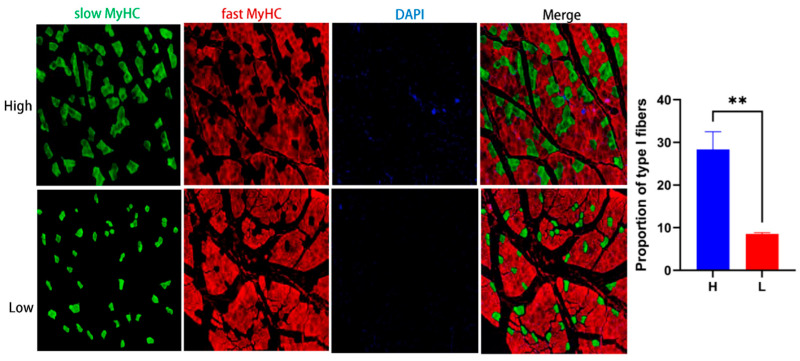
Immunofluorescence images of *longissimus lumborum* muscle in lambs with high vs. low proportions of slow-twitch fibers. Representative images and quantification of slow-twitch (type I) fibers from the high and low groups (*n* = 6). Slow muscle fibers are stained with green fluorescence, fast muscle fibers are stained with red fluorescence, and the cell nuclei are blue. Bar graphs show mean ± standard error (SEM); ** indicates *p* < 0.01.

**Figure 6 foods-14-02309-f006:**
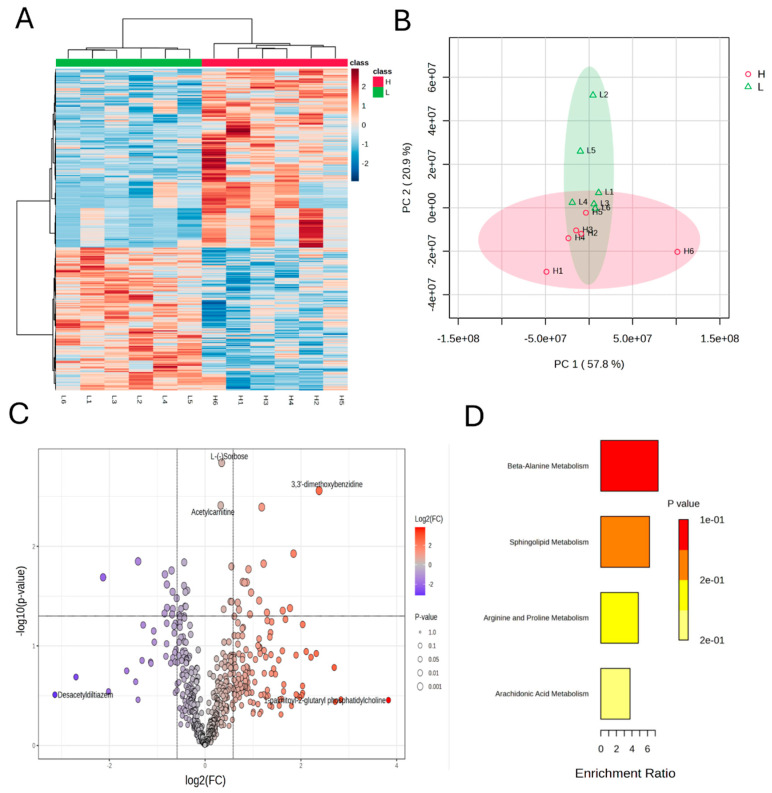
Bioinformatics analysis of metabolomic data. (**A**) Hierarchical clustering heatmap of differential metabolites. (**B**) Principal component analysis (PCA) plot of metabolite profiles. The 95% confidence intervals for each group are represented by ellipses, with the H group in pink and the L group in green. (**C**) Volcano plot showing significantly up- and downregulated metabolites between groups. (**D**) KEGG pathway enrichment analysis of differential metabolites.

**Table 1 foods-14-02309-t001:** Composition and nutrient levels of basal diets (air-dry basis, %).

Items	Value
Diet Composition	
Leymus chinensis, % DM	50.0
Corn, % DM	20.0
Soybean meal, % DM	5.4
DDGS, % DM	7.0
Soybean oil, % DM	15.0
CaHPO_4_, % DM	0.5
Limestone, % DM	1.4
NaCl, % DM	0.5
Vitamin + Mineral additives, % DM	0.2
Nutrient Level	
ME, MJ/kg	11.99
CP, % DM	10.1
Ca, % DM	0.65
TP, % DM	0.33

Note: The mineral and vitamin formulation per kilogram of feed included the following: vitamin A: 30,000–200,000 IU, vitamin D: 30,000–200,000 IU, vitamin E: 2000–30,000 IU, iron: 500–2400 mg, cobalt: 2.0–16.0 mg, copper: 100–300 mg, zinc: 400–2400 mg, manganese: 400–2000 mg, iodine: 2.0–10.0 mg, and selenium: 2.0–8.0 mg. Nutrient levels are expressed as percentages.

**Table 2 foods-14-02309-t002:** Primer sequences used for qPCR.

Target Gene	Primer Sequence (5′ → 3′)	Product Size (bp)	Annealing Temperature (°C)
*MyHC Ⅰ*	F: GAACAGGCCAACACCAACCT	145	60
	R: CCTCATTCAAGCCCTTGGCG		
*MyHC Ⅱa*	F: AGTATGAGGAAACCCACGCC	159	60
	R: GGTCAGAGATCTCCTGCTGC		
*MyHC Ⅱb*	R: AGGCTGTCAAAGGTCTTCGGAAAC	81	60
	R: GAACATTCTTGCGGTCCTCCTCAG		
*MyHC Ⅱx*	F: GCTCCACCTGGATGATGCTCTG	140	60
	R: TCTTCCTGCTCCTCTCCGTCTG		
*β-actin*	F: TTCTAGGCGGACTGTTAG	84	60
	R: TGCCAATCTCATCTCGTT		

**Table 3 foods-14-02309-t003:** Expression levels of MyHC isoforms.

Item	Sample Size	Minimum	Maximum	Mean	Standard Error
MyHC I	30	0.01	2.75	1.13	0.73
MyHC IIx	30	0.3	2.2	0.823	0.36
MyHC IIa	28	0.75	1.61	1.223	0.26
MyHC IIb	30	0.05	2.02	0.993	0.45

**Table 4 foods-14-02309-t004:** Descriptive statistics of meat quality traits.

Item	Sample Size	Minimum	Maximum	Mean	Standard Error
Shear force (N)	30	26.61	85.92	48.02	2.69
pH_24_	30	5.36	6.94	6.012	0.07
Cooking loss(%)	30	54.78	64.83	57.49	0.397
L*	30	30.99	43.18	37.06	0.647
a*	30	12.03	18.53	14.79	0.316
b*	30	7.05	16.33	12.72	0.44
Moisture(%)	30	68.03	72.74	70.40	0.19
Protein(%)	30	21.37	23.63	22.37	0.10
Fat (%)	30	2.42	6.31	4.29	0.19

**Table 5 foods-14-02309-t005:** Fatty acid composition (% of total fatty acids) in the *longissimus lumborum* muscle of lambs.

Fatty Acid (% of Total Fatty Acids)	Sample Size	Minimum	Maximum	Mean	Standard Error
C14:0	30	1.11	2.55	1.84	0.06
C15:0	30	0.00	0.29	0.16	0.01
C15:1	30	0.00	0.22	0.03	0.01
C16:0	30	19.80	27.00	24.32	0.29
C16:1	30	1.25	2.17	1.68	0.04
C17:0	30	0.53	1.21	0.80	0.02
C17:1	30	0.45	1.02	0.60	0.02
C18:0	30	14.90	22.60	18.15	0.36
C18:1n-6	30	39.80	50.30	45.21	0.42
C18:2 n-6	30	3.35	8.82	5.42	0.25
C20:4 n-6	30	1.00	3.13	1.72	0.11
SFA	30	40.49	50.40	45.30	0.42
UFA	30	49.52	59.52	54.69	0.42
MUFA	30	41.56	52.64	47.53	0.44
PUFA	30	4.54	11.83	7.15	0.344

Note: C14:0, myristic acid; C15:0, pentadecanoic acid; C15:1, cis-10-pentadecenoic acid; C16:0, palmitic acid; C16:1, palmitoleic acid; C17:0, heptadecanoic acid; C17:1, cis-10-heptadecenoic acid; C18:0, stearic acid; C18:1n-6, gamma-linolenic acid; C18:2 n-6, linoleic acid; C20:4 n-6, arachidonic acid; SFA, saturated fatty acids; UFA, unsaturated fatty acids; MUFA, monounsaturated fatty acids; PUFA, polyunsaturated fatty acids.

**Table 6 foods-14-02309-t006:** Amino acid composition in the *longissimus lumborum* muscle of lambs.

Amino Acids (g/100 g)	Sample Size	Minimum	Maximum	Mean	Standard Error
Aspartic Acid	30	5.53	6.30	5.93	0.03
Threonine	30	2.86	3.23	3.05	0.021
Serine	30	2.29	2.66	2.49	0.02
Glutamic Acid	30	10.09	11.53	10.79	0.07
Proline	30	1.42	1.82	1.59	0.02
Glycine	30	2.60	2.97	2.77	0.02
Alanine	30	3.40	3.93	3.65	0.02
Valine	30	2.83	3.41	3.07	0.02
Methionine	30	1.17	1.49	1.36	0.01
Isoleucine	30	2.68	3.16	2.91	0.02
Leucine	30	4.89	5.61	5.25	0.03
Tyrosine	30	2.09	2.37	2.25	0.01
Phenylalanine	30	2.27	2.91	2.74	0.02
Histidine	30	2.48	3.07	2.85	0.02
Lysine	30	5.14	6.16	5.67	0.04
Arginine	30	3.70	4.35	4.01	0.03
EAA	30	25.17	28.83	26.53	0.15
UAA	30	32.13	35.69	34.20	0.12

Note: EAA, essential amino acids; UAA, non-essential amino acids.

**Table 7 foods-14-02309-t007:** Volatile flavor compound contents in the *longissimus lumborum* muscle of lambs.

Compounds (μg/mL)	Sample Size	Minimum	Maximum	Mean	Standard Error
Hexanal	30	0.00	0.80	0.52	0.03
Benzaldehyde	30	0.00	0.42	0.35	0.01
1-Pentene-3-ol	27	0.01	0.03	0.02	0.00
2,4-Decadienal	23	0.40	0.41	0.40	0.00
Nonanal	28	0.83	2.22	1.20	0.06
4-Methyloctanoic acid	27	1.72	4.29	1.93	0.10
4-Methylnonanoic acid	29	2.21	5.36	2.44	0.11

**Table 8 foods-14-02309-t008:** Differential metabolites in the *longissimus lumborum* muscle of lambs with different proportions of slow-twitch fibers.

No.	Differential Metabolites	log_2_(FC)	*p*-Value
1	3,3′-dimethoxybenzidine	2.3796	0.0027683
2	C17-sphinganine	1.1807	0.0040576
3	Isobutyryl-l-carnitine	1.8451	0.011826
4	2-propanoyloxy]propyl]oxolan-2-yl]propanoic acid	−1.3998	0.014139
5	Carbendazim	1.2198	0.014916
6	2-methylbutyryl-l-carnitine	0.90414	0.016988
7	1-Myristoyl-sn-glycero-3-phosphocholine	−0.70102	0.017468
8	Thr-His	−0.83994	0.019106
9	Met-Phe	−2.1315	0.020461
10	4,4′-methylenebis(2,6-di-tert-butylphenol)	0.78419	0.022737
11	Juglone	0.86652	0.022992
12	Evodiamine	0.82165	0.023152
13	Phosphocreatine	−0.79908	0.024145
14	Ala-Ile	−0.67899	0.028617
15	N-2-hydroxyethylpiperazine-n-3-propanesulfonic acid	0.95851	0.029074
16	1-hexadecenyl-2-acyl-sn-glycero-3-phosphocholine	−0.59397	0.033177
17	L-tryptophanamide	1.1274	0.035059
18	N-arachidonoyldopamine	0.67904	0.0367
19	5-methylcytidine	−0.79492	0.041314
20	Lys-Trp-Arg	1.7708	0.041758
21	Ile-Met	−0.67514	0.042485
22	Tomatidin	0.70509	0.043277
23	Gamma-glutamylvaline	1.6194	0.044228
24	2′-hydroxy-2,5,6′-trimethoxychalcone	1.2899	0.046209
25	Methylergonovine	−0.84761	0.04724
26	His-Val	0.93539	0.048256
27	Asn-Asn	−0.68984	0.04842

## Data Availability

The original contributions presented in this study are included in the article. Further inquiries can be directed to the corresponding authors.

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
