# Peer review of "Effects of Muscle Fiber Composition on Meat Quality, Flavor Characteristics, and Nutritional Traits in Lamb"

_foods, 2025, doi:10.3390/foods14132309_

Round 1

Reviewer 1 Report

Comments and Suggestions for Authors

An interesting paper on lamb meat fiber types and eating quality.  Some comments below:

L35.  I see "mutton" is a key word.  The study is not about mutton.  Is there a specific reason the authors chose this as a key word?

L67 and throughout.  The name of the muscle should be in italics.

L110.  The authors state that male lambs are used.  Were these intact males or castrates?  This would have a huge impact on the applicability of this research.  It should be stated.

L134.  The authors state that a calibrated pH meter was used but not how it was used.  Was the probe a spear-tip probe or was the muscle sample homogenized in water.  Please be a bit more precise.

L139.  The authors did a good job of describing the color measures.  Was the spectral component included or excluded?

L142.  What sort of texture analyzer was used?  How was the instrument set up?  Who made it?

Table 2.  Moisture is misspelled.

L 324 & L332.  This is another Table 2.  This is confusing. Is the table at L332 supposed to be Table 6?  Also, in the title for L332, I am thinking it may be advisable to include that the identity of these metabolites was derived from the volcano plot.  This may help tie the information together.

L334.  The enrichment ratio in Figure 6D is hard for me to interpret.  It just looks like two colored boxes that have little meaning on their own.  I would suggest the authors take more care in explaining what this means or leave it out.  

L377.  The authors are reiterating their findings here.  There is a lot of meat science wrapped up in this sentence that has not been addressed in the paper.  This includes things like chilling rate, cold induced toughening, etc.  These things have been left unsaid.  But perhaps that is fine in the course of this paper.  It is hard to include all aspects of meat science in one paper.

There seems to be many authors for this paper.  Please be careful on assigning authors to those who actually make significant contributions to it and take responsibility for it.  Of course, the determination of these is up to the institution's guidelines.

Author Response

Response to Reviewer Comments

Submission ID: foods-3724085

Title: “Effects of Muscle Fiber Composition on Meat Quality, Flavor Characteristics, and Nutritional Traits in Lamb”

We sincerely thank you for your valuable feedback and constructive suggestions on our manuscript(foods-3724085). We have carefully checked and revised the manuscript according to your comments. Please find my itemized responses and my revisions and corrections in the re-submitted files below. The reviewer comments are given in normal font. Our response is given in the blue text and changes/additions to the manuscript are given in the red text.

We hope the revised manuscript will meet your journal’s standard. We greatly appreciate your time and effort to improve our manuscript for publication.

Best wishes,

Fu Yu

L35.  I see "mutton" is a key word.  The study is not about mutton.  Is there a specific reason the authors chose this as a key word?

Thank you for your attention to our study and your valuable feedback. Regarding your comment about the keyword "mutton," we would like to clarify that "mutton" typically refers to meat from adult sheep, whereas our study focuses on 6-month-old lambs. In light of this, we have decided to remove "mutton" as a keyword to more accurately reflect the content of our research.  

L67 and throughout.  The name of the muscle should be in italics.

Thank you for your insightful comments. We appreciate your attention to detail regarding the formatting of muscle names. we have made the necessary changes to italicize the name of the muscle throughout the manuscript, including on line 67.

L110.  The authors state that male lambs are used.  Were these intact males or castrates?  This would have a huge impact on the applicability of this research.  It should be stated.

Thank you for your important question regarding the use of male lambs in our study. We would like to clarify that we used intact male lambs that were 6 months old. We have added this information to the materials section of the manuscript to ensure clarity and address your concern regarding the applicability of our research. Thank you for your feedback, which has helped us improve our paper.

L134.  The authors state that a calibrated pH meter was used but not how it was used.  Was the probe a spear-tip probe or was the muscle sample homogenized in water.  Please be a bit more precise.

Thank you for your insightful comment regarding the measurement of pH in our study. To clarify, the ultimate pH (pH24) of the longissimus lumborum muscle was determined 24 hours postmortem using a calibrated pH meter (S220, Mettler Toledo) with a spear-tip probe directly inserted into the muscle sample. This detail is included in the revised manuscript for better precision.

L139.  The authors did a good job of describing the color measures.  Was the spectral component included or excluded?

Thank you for your valuable feedback. We have revised the manuscript to include the clarification that the spectral component was excluded from our color measurements. This detail has been added to ensure a clearer understanding of our methodology. We appreciate your guidance in enhancing the clarity of our work.

L142.  What sort of texture analyzer was used?  How was the instrument set up?  Who made it?

Thank you for your valuable comment. We have revised the Methods section to provide detailed information about the instrument used for texture analysis. Specifically, we added the instrument model, manufacturer, shear blade type, sample preparation procedure, and testing parameters.

Table 2.  Moisture is misspelled.

Thank you for catching this oversight. The spelling of “Moisture” has been corrected in the revised manuscript.

 L334.  The enrichment ratio in Figure 6D is hard for me to interpret.  It just looks like two colored boxes that have little meaning on their own.  I would suggest the authors take more care in explaining what this means or leave it out. 

Thank you for your valuable comment. We agree that the previous version of Figure 6D lacked clarity. In the revised manuscript, we have adjusted the plotting parameters to enhance the readability of the enrichment analysis results. The updated figure now includes clearer visual indicators (e.g., enrichment ratio scale, pathway names, and corresponding P-values). We have also revised the corresponding description in the Results and Discussion sections to provide a more accurate and interpretable explanation of the enriched pathways and their biological implications.

L377.  The authors are reiterating their findings here.  There is a lot of meat science wrapped up in this sentence that has not been addressed in the paper.  This includes things like chilling rate, cold induced toughening, etc.  These things have been left unsaid.  But perhaps that is fine in the course of this paper.  It is hard to include all aspects of meat science in one paper.

Thank you for your thoughtful observation. We agree that factors such as chilling rate and cold-induced toughening are important contributors to postmortem meat quality. However, given the scope of this study—which primarily focuses on muscle fiber type, MyHC isoform expression, and their associations with meat quality traits including flavor, metabolomics, and nutritional composition—we chose not to elaborate on these postmortem processing variables to maintain focus and clarity. We appreciate your understanding that it is difficult to address every aspect of meat science in a single manuscript, and we believe this decision helps to retain a clear narrative centered on our core findings.

There seems to be many authors for this paper.  Please be careful on assigning authors to those who actually make significant contributions to it and take responsibility for it.  Of course, the determination of these is up to the institution's guidelines.

We appreciate the reviewer’s reminder regarding authorship responsibilities. We confirm that all listed authors have made substantial contributions to the conception, design, execution, or interpretation of the study and have approved the final version of the manuscript. The assignment of authorship strictly followed our institutional and journal guidelines to ensure fairness and accountability. Thank you again for your attention to this important aspect of scientific publishing.

Reviewer 2 Report

Comments and Suggestions for Authors

The manuscript is an interesting contribution in the sense of studying the correlation between muscle myosin HC type and physicochemical and sensory attributes of lamb. The manuscript has scientific merit. I have some considerations for the authors.

110-112: Please provide a better description of the feeding and management conditions of the animals, since feeding can also be a factor.

124-125: Could the authors provide these results (expression levels)?

133: pH24 I think.

240-241: The authors should provide a deeper explanation on why MyHC1 I is related to UFA in the Discussion section. It is not clear.

287-288: Is this correlation negative or positive? It is closer to 1.0 than -1.0.

In general, the Discussion must be improved.

My recommendation is Major revision.

Author Response

Response to Reviewer Comments

Submission ID: foods-3724085

Title: “Effects of Muscle Fiber Composition on Meat Quality, Flavor Characteristics, and Nutritional Traits in Lamb”

We sincerely thank you for your valuable feedback and constructive suggestions on our manuscript(foods-3724085). We have carefully checked and revised the manuscript according to your comments. Please find my itemized responses and my revisions and corrections in the re-submitted files below. The reviewer comments are given in normal font. Our response is given in the blue text and changes/additions to the manuscript are given in the red text.

We hope the revised manuscript will meet your journal’s standard. We greatly appreciate your time and effort to improve our manuscript for publication.

Best wishes,

Fu Yu

110-112: Please provide a better description of the feeding and management conditions of the animals, since feeding can also be a factor.

Thank you for your insightful suggestion. We have revised the “Materials and Methods” section to include more detailed information about the feeding and management conditions. Specifically, we now state that all lambs were housed indoors (stall-fed) under standardized management conditions, with ad libitum access to water and a balanced diet formulated to meet their nutritional requirements. The composition and nutritional content of the feed are provided in Table 1.

124-125: Could the authors provide these results (expression levels)?

Thank you for your request regarding the expression levels of MyHC isoforms. We have added this information to the manuscript and included it in Table 3, titled "Expression Levels of MyHC Isoforms." We appreciate your suggestion, which has helped improve the completeness of our study.

133: pH24 I think.

Thank you for pointing out the error regarding pH42. We have corrected this to pH24 in the manuscript. We appreciate your attention to detail, which has helped us improve the accuracy of our work.

240-241: The authors should provide a deeper explanation on why MyHC1 I is related to UFA in the Discussion section. It is not clear.

Thank you for your valuable comment. We have expanded the Discussion section to clarify the mechanistic basis for the observed association between MyHC I expression and unsaturated fatty acids (UFAs). Specifically, we explain that oxidative (slow-twitch) muscle fibers, characterized by MyHC I expression, possess greater mitochondrial density and enzymatic activity related to fatty acid β-oxidation, which contributes to the preferential accumulation and utilization of unsaturated fatty acids. These physiological properties of oxidative fibers help explain the observed enrichment of UFAs in MyHC I-rich muscles.

287-288: Is this correlation negative or positive? It is closer to 1.0 than -1.0.

Thank you for your insightful question regarding the correlation. We acknowledge the error in our previous statement. The heatmap shows that MyHC IIx expression exhibited a significant negative correlation with 1-penten-3-ol (p < 0.05), while MyHC IIa demonstrated a significant positive correlation with 1-penten-3-ol (p < 0.01). We have made the necessary modifications in the manuscript. We appreciate your guidance in improving the clarity and accuracy of our work.

In general, the Discussion must be improved.

 We sincerely thank the reviewer for this important suggestion. In response, we have carefully revised and expanded the Discussion section to improve its clarity, depth, and logical structure. Specifically, we have:

Expanded mechanistic explanations – We provided a clearer rationale for the observed associations, particularly explaining why oxidative MyHC I fibers are positively correlated with unsaturated fatty acids (UFAs), by linking this to known differences in mitochondrial density, β-oxidation capacity, and oxidative metabolism.

Incorporated relevant literature – We strengthened the scientific grounding of our discussion by referencing additional studies in pigs, cattle, and other species that support or contrast with our findings.

Clarified novel insights – We highlighted and explained new observations, such as the differential effects of MyHC IIa and IIx on specific volatile flavor compounds like 1-penten-3-ol, and the implications of intermediate fiber types in modulating flavor profiles.

Refined logical flow – We reorganized the Discussion to follow a clearer progression from technological traits, nutritional traits, flavor traits, to metabolomics, culminating in practical implications for meat production and quality control.

We hope that these comprehensive revisions have significantly improved the scientific value and readability of the Discussion section.

Round 2

Reviewer 2 Report

Comments and Suggestions for Authors

The authors addressed my comments and questions. I have no more comments.